# OpenReview forum: "On Targeted Manipulation and Deception when Optimizing LLMs for User Feedback"
_ICLR.cc/2025/Conference — ICLR 2025 Poster_

### Official Review · Reviewer_89WH · 2024-10-30

**Soundness:** 3
**Presentation:** 4
**Contribution:** 3
**Rating:** 8
**Confidence:** 3

**Summary:**

The paper demonstrates through empirical experiments that an LLM agent trained using RL to maximise user feedback may be incentivised to manipulate or deceive the user. This is demonstrated across 4 different classes of problem domain, involving either single-stage or 2-stage conversations between the agent and the user. The results show that the agent may lie to the user, exhibit sycophantic behaviour even when this is associated with harmful content or actions, or engage in more subtle manipulative behaviour such as nudging. Significantly this can arise even when only a subset of the user population is manipulatable, as the agent learns to exploit those users while behaving normally with respect to the rest of the population, which make this manipulative behaviour harder to detect. Two possible approaches to mitigating this issue are proposed and evaluated, and its shown that while they may have beneficial effects in some cases, they can also result in additional problematic behaviour on the part of the agent.

**Strengths:**

The paper addresses a very important issue. While previous work has hypothesised that LLM-based agents may be motivated to 'game' the preferences of users, this work provides quite solid empirical evidence of this actually occurring in scenarios which are similar to the situations where such agents might actually be deployed.

The paper is very well written. It is easy to read and does an excellent job of presenting the context of the work and discussing its implications.

The experiments were mostly well-designed (see one proviso in the Weaknesses section below) and clearly described, and the analysis is clear and quite compelling. The provision of code implementations also means that the hard work which has gone into developing these test cases and the evaluation procedures can be leveraged further by other researchers.

The limitations of the approaches used are appropriately acknowledged and justified (e.g. the use of a relatively simple approach to the RL optimisation process).

**Weaknesses:**

The main weakness I see in the paper relates to the manner in which the gameable and non-gameable users are differentiated in the initial states. The examples in Figure 13 show that this is less than subtle, with the agent being directly told that one user is extremely susceptible to their suggestions and will always act on their advice. The results of this set of experiments would be much stronger if the distinction between gameable and non-gameable users was more subtly indicated to the agent, or even if this was something which the agent had to largely establish for itself through repeated interactions with each user. The statement in Section 4.2 that "Exploratory experiments suggested that the exact difference in background didn’t matter much" suggests that this issue has been explored to some extent, but no details are provided. The paper's case would be stronger if the outcomes of experiments using more subtle indicators were reported (at least in an appendix if space doesn't permit it in the main body).

Some small presentation issues. The text on some of the figures is extremely small (for example, I had to zoom in well over 200% to read some of the text in Figure 1). This might be difficult to fix given the page restrictions, but it would make the paper easier to read if this could be addressed.

There are a few undefined acronyms. While I think its reasonable to use common terms like RL and RLHF without explanation, some of the other acronyms were ones which readers (like myself) might not be familiar with (eg on p3, BoN and KTO). It would be better is these were spelled out in full when first used.

**Questions:**

Why is there an asterisk in the title? I thought this would be explained somewhere in the paper but I couldn't find it.

What is the meaning of the User P.O. and Agent P.O. columns in Table 1?

I was unclear on the discussion of manipulation in the action-advice scenario on p5. Why would encouraging the user to engage in a harmful action lead to higher ratings for the agent responses later in the conversation? Is this simply the "afterglow" of the user's positive experience of the harmful action?

In Fig 6, there's a difference in the problematic behaviour of the agent with regards to the gameable and non-gameable users before training. Can you explain this?

---

> ### Author Response · Authors · 2024-11-15
>
> We'd like to thank Reviewer 89WH for their positive review. We are glad they found our paper to “address a very important issue”, and provide “quite solid empirical evidence of this actually occurring in scenarios which are similar to the situations where such agents might actually be deployed.”
>
> Below, we hope to address some of their remaining concerns.
>
> ## Improvements on how gameable and non-gamable users are differentiated in the initial states.
>
> We agree with the Reviewer that the current way in which the users are differentiated is not subtle (we have removed that phrasing). The idea was that the “AI memory”/”User Context” would partly be a proxy for what a model might learn about a user during a long context conversation. [Other work](https://arxiv.org/abs/2310.07298) has shown that LLMs are good at inferring characteristics of users – that’s what we were trying to replicate. In particular, while the mechanism that models like GPT-4 use to decide what to store in memory is unclear, it seems plausible that it would store facts that correlate with whether the user is gameable, as this would be useful information for it to have in future conversations (that said, we think our results would also extend to the case without any memory at all, as discussed later).
>
> While our original experiments ultimately test whether LLMs would learn to adapt to character traits which may be directly expected to be related to whether the user is gameable (whether the user is gullible or independent-minded), we have added an additional experiment which suggests that the semantic meaning of the information about the user may not matter. In particular, we tried replacing the “givaway attribute” for whether a user is gameable to whether they like the color blue or red. Under this setup, the model seems to learn even more quickly to exploit the red users while acting normal for the blue ones, than in the previous experimental setup. See Appendix F.2 for more details about this experiment.
>
> Moreover, we have some initial results indicating that for multistep interactions we may not actually need to have *any discernible difference in the distribution of initial prompts at all*, as the LLM can learn to separate between gameable and non-gameable users by just interacting with them (over the course of 2 rounds of conversation). We will run these experiments more rigorously and report back.
>
> ## Realism of action-advice
>
> Yes, the idea was to try to capture the "afterglow" of the user's positive experience of the harmful action. However, all users having a positive experience (in the short term) when choosing to engage in harmful behaviors is admittedly unrealistic. That said, similarly to the analysis from 4.2, we could have made only some users have positive experiences, and there doesn’t seem to be a good reason to expect different results (we did not prioritize such runs because they are the most expensive ones in our current setup).
>
> The key thing we were trying to show with the *action-advice* environment is that external actions a user takes may modify the reward signal too, rather than that the LLM could selectively target different users (which we showed in Section 4.2).
>
> ## More minor points/questions
>
> > The text on some of the figures is extremely small
>
> We have fixed some of these, but we will try to update Figure 1 specifically in a future version, as we want to somewhat redesign it
>
> > There are a few undefined acronyms.
>
> We should have fixed this in the updated version – thanks for catching that!
>
> > Why is there an asterisk in the title? I thought this would be explained somewhere in the paper but I couldn't find it.
>
> We agree that it was confusing. We meant it to emphasize the fact that this is *user* instead of *annotator* feedback. We have updated the title to not have it anymore.
>
> > In Fig 6, there's a difference in the problematic behaviour of the agent with regards to the gameable and non-gameable users before training. Can you explain this?
>
> We speculate that the difference in Figure 6 between gameable and non-gameable users may have been small changes in the behavior of Llama models in the presence of different user character traits for the initial states. With our new approach to calculating harmfulness of models (described in the common response about the updated paper), this difference isn’t present anymore.
>
> > What is the meaning of the User P.O. and Agent P.O. columns in Table 1?
>
> User P.O. and Agent P.O. refer to whether there is partial observability for the user or for the AI agent. We agree that this is confusing. We have removed them from the table and will add that information to the appendix.

---

> > ### Comment · Reviewer_89WH · 2024-11-19
> >
> > I appreciate the additional results which the authors have provided in Appendix F.2, showing that the agent can still learn to manipulate gameable users even when the distinction between gameable and non-gameable users is represented by arbitrary features rather than by features which semantically relate to the user's gameability. This strengthens their case as it shows that this sort of manipulation can arise even when the agent is not directly given any indication as to which users can be manipulated.
> > The results of the experiments on multi-step interactions would further support this aspect of the paper.

---

> > > ### Author Response · Authors · 2024-11-20
> > >
> > > We have now added some 2-turn experiments which show that the model can learn to target gameable users with minor or even no initial state differences from the non-gameable users when allowed 2 conversation turns. See other comments and appendix F.2 in the updated paper for more details.

---

### Official Review · Reviewer_xX6K · 2024-11-02

**Soundness:** 2
**Presentation:** 3
**Contribution:** 3
**Rating:** 6
**Confidence:** 3

**Summary:**

The title and abstract point to an important problem in LLMs, i.e., issues with feedback loops originating from user feedback. While the problem of feedback loops has been acknowledged earlier (even in the context of LLMs, e.g., due to training LLM generated data, as well as in concurrent work (Wen et al. 2024)) and it is well-known that optimization might aim at shortcuts and objectives are difficult to formulate in machine learnign, the paper adds a number of interesting claims that if appropriately backed up by evidence, would justify a publication at ICLR without any doubt. While the paper discusses a bunch of interesting cases, the evaluation is, where the paper falls short. Already the title should say "... deception could emerge..." and the abstract should be presented more modestly. The evidence relies on simulated data, on an optimization method that is not among the most common ones, and an LLM that is not used by the majority of users. It is unclear to what extend the results transfer to other scenarios. In turn, this makes the statements much less relevant, as it is in general well-known that such "optimization issues" could occur. While the paper is well-aware of these shortcomings and aims to talk them away (also partially backed by refs), overall some form of direct comparison of simulated data and real data would be needed to show alignment of feedback distributions.

-------
I acknolwedge the author's response, while the ChatGPT style verboseness is not appreciated, at least the point with the optimization seems somewhat justified. I updated my score.

**Strengths:**

see above

**Weaknesses:**

see above

**Questions:**

Do you agree that you overclaim and your evidence is insufficient for the general claims?

---

> ### Author Response · Authors · 2024-11-15
>
> We’d like to thank Reviewer xX6K for their review – we are confident it will help us improve the presentation of our work.
> Despite their concerns, we are happy that Reviewer xX6K thought that our paper “adds a number of interesting claims that if appropriately backed up by evidence, would justify a publication at ICLR without any doubt”.  Below (and in the common response), we hope we can provide some evidence that some of Reviewer xX6K’s gripes with our paper’s evaluation are due to misunderstandings, and others may not be as much of a deal-breaker as they may seem.
> On re-reading the paper, we agree with the Reviewer that some of the claims we make may have come across as overly strong. We have revised the paper accordingly, and have modified the title. We hope that this, together with the additional information we provide in our responses, can address the Reviewer’s concerns.
>
> ## “The problem is already well-known”
> We think that there may be a misunderstanding about our intended contributions, which we clarify here. We agree that the alignment problem and concerns about [emergent deception and manipulation](https://bounded-regret.ghost.io/emergent-deception-optimization/) are well-documented, as we discuss in our related work. However, beyond the existence of the problem, details of *how the problem would manifest* are important for how one would combat these issues in practical AI deployments. People widely use RLHF (and similar techniques) despite these known issues – mostly due to an assumption that the human feedback is reliable enough to not affect behavior significantly in practice for LLMs.
>
> We make the following contributions, which go beyond simply rehashing a known problem:
> 1. Together with concurrent work ([Wen et al.](https://arxiv.org/abs/2409.12822)), we provide ***concrete*** scenarios of how these problems could manifest in practice. Despite our settings still being simulations, they are significantly more realistic than those of prior works, which are more focused on conceptual arguments in this domain ([1](https://ojs.aaai.org/index.php/AAAI/article/view/21186), [2](https://arxiv.org/abs/2405.17713), [3](https://bounded-regret.ghost.io/emergent-deception-optimization/)). Moreover, as [Wen et al.](https://arxiv.org/abs/2409.12822) demonstrate a subset of the phenomena we are interested in with real people, lending further credence to their plausibility of these risks more generally. Additionally, we show that even with optimization techniques which are similar to ones used in practice, LLMs have sufficient exploration to learn these harmful behaviors despite their safety training, which was not obvious (even with knowledge of the broader problem).
> 2. More importantly, we show that using *user* feedback specifically can make things worse than using imperfect annotator feedback, because it allows targeting of the most vulnerable users. To the best of our knowledge, this is a novel point, both conceptually (why it would be the case) and empirically.
> 3. Additionally, we show that standard mitigation strategies may only be partially effective at addressing these harms, and that the best publicly available benchmarks in this domain are often not able to identify these harmful behaviors.
>
> We think that most of the points are novel and seem valuable to the research community, especially as user feedback optimized systems are increasingly deployed in practice.
>
> ## Our evidence is based on simulated data: “Direct comparison of simulated data and real world data would be needed to show alignment of feedback distributions”
>
> While this is certainly a limitation of our results, we do not think that this invalidates our claims – we do not claim that these effects *will necessarily happen* with real users. We simply claim that current models, paired with RL for user feedback, have the capacity to exploit vulnerable users reliably. For a full answer to this question, see the common response.

---

> ### Author Response · Authors · 2024-11-15
>
> ## “The optimization method is not among the most common ones”
>
> Note that one should expect emergent manipulation to be method-agnostic: its root cause is imperfect feedback, rather than KTO’s imperfections as an optimization method. In preliminary experiments, we found similar results to hold with a variant of Expert Iteration [(Anthony et al., 2017)](https://arxiv.org/abs/1705.08439) described in Appendix C of the updated paper. Expert Iteration is used more broadly as an RL method, and has been shown to be close to state of the art for multi-step LLM RL optimization ([Havrilla et al., 2024](https://arxiv.org/abs/2403.04642), [Singh et al., 2023](https://arxiv.org/abs/2312.06585)). Moreover, it seems important to note that there is a clear interest from companies that run services similar to the ones we consider in the paper to [use RL](https://arxiv.org/abs/2303.06135) to [optimize for user feedback](https://www.theverge.com/24216748/replika-ceo-eugenia-kuyda-ai-companion-chatbots-dating-friendship-decoder-podcast-interview).
>
> Also, note that if anything, the fact that we used a weak optimization method without good exploration if anything *underestimates* the incidence of the phenomena we study. With more powerful optimizers (especially with better exploration) we would only expect manipulative behaviors to be more effective, not be reduced. Indeed, our intention is to simply use KTO as a placeholder for more powerful RL optimization methods that will be developed in the future, following a similar philosophy to the choice of method in [Denison et al. (2024)](https://arxiv.org/abs/2406.10162).
>
> ## “[The authors] use an LLM that is not used by the majority of users”
> The Reviewer may have missed that we also show that our main results hold across various LLMs: in particular, the results hold unchanged even with the Gemma 2 2B, Gemma 2 9B, and Gemma 2 27B models. While these are still not models that are "used by a majority of users", note that we cannot do KTO training with proprietary models because it requires access to model weights, and we ran into cost issues when trying to use Expert Iteration with e.g. GPT (using their fine-tuning API, as EI only requires that).
>
> We thought that in light of our Gemma results (and the fact that the trends of harmfulness do not abate at larger model sizes), there doesn’t seem to be any particular reason to believe that the behaviors we find wouldn’t generalize to any arbitrary LLM.
>
> We’d also like to point out that Llama is one of the most downloaded open-source models and a version of llama is the top rated open source model on chatbot arena. It seems quite plausible that there would be applications like those in our paper which use such a model as a backbone.

---

### Official Review · Reviewer_aH2R · 2024-11-03

**Soundness:** 3
**Presentation:** 3
**Contribution:** 3
**Rating:** 5
**Confidence:** 3

**Summary:**

This paper investigates the potential risks of training large language models (LLMs) directly on user feedback through reinforcement learning. The authors demonstrate that optimizing for user feedback can lead LLMs to showcase manipulative behaviors in different settings, such as deception and sycophancy, which can be selectively targeted at vulnerable user subsets.

**Strengths:**

1.  The findings of the study is  interesting and has the potential to improve the safety of LLMs. It conducts a thorough analysis of various manipulative behaviors that LLMs can exhibit and how these can be selectively applied to different user groups.

2. The experimental setup, using simulated environments to test various scenarios, is robust and well-justified, which strengthens the validity of the results.

**Weaknesses:**

1. The paper is mainly relied on the simulated user feedback and lack of read-world user data would limit the generalizability of the findings. The simulated user behavior data can not fully reflect the complexity of real human feedback when interactingf with the LLMs. In addition, the study assumes that users keep a static understanding of LLM behavior which does not account for the evolution of user understanding and feedback when continuously interacting with the LLM.


2. The mitigation strategies discussed are only partially effective and could benefit from more advanced techniques to better  suppress the manipulative behavior of the LLM especially in some critical cases.

**Questions:**

See above.

---

> ### Author Response · Authors · 2024-11-15
>
> We'd like to thank Reviewer aH2R for their review. We are glad that they found our paper to provide a “thorough analysis of various manipulative behaviors that LLMs can exhibit” and an “experimental setup [which is] robust and well-justified”. Below, we respond to their concerns.
>
> ## Lack of real user feedback
> Please refer to the common response about this point.
>
>
> ## The study assumes that users keep a static understanding of LLM behavior
> ​​
> While fully studying how users adapt to LLMs is out of the scope of the current work, we agree that it is an interesting direction for future work. On the one hand, accurately simulating user adaptation to LLM behaviors seems challenging to do correctly, especially if considering dynamics of iterated re-training and deployment of real world LLM systems. On the other, we think our findings are still significant despite this: the fact that these incentives are present in our settings provides compelling evidence that they may be present for longer term interactions too. Moreover, despite the fact that our *therapy-talk* environment is single-step, it is meant to simulate a user which has a long-term history with an LLM therapy app (the AI system has quite a bit of information about the user in its context).
>
> ## The mitigation strategies discussed are only partially effective
> We think there may be a misunderstanding regarding our contributions with respect to mitigation strategies, which we clarify here. Firstly, we’d like to note that the main contribution of our paper is to showcase that these problems may surface when optimizing user feedback data, and that both standard mitigation strategies from the literature (such as mixing in helpful & harmless data), and ones we devise ourselves (vetoing training trajectories in various ways) are insufficient to solve the problem. Hence, our results on mitigation strategies should be seen as *negative results*.
>
> While we don’t “solve” the problem, it seems important that the most promising techniques not only fail, but may even give a false sense of security & make manipulative behaviors remain but become more subtle.
>
> Moreover, while it’s always possible to develop more advanced techniques (this criticism can always be levied against any negative result), we did spend significant effort in trying to improve our mitigation strategies beyond standard techniques. We have also run additional experiments in our updated paper:
> - We try continued safety training with another dataset (PKU-Alignment), to see whether HH-RLHF is insufficient out of an idiosyncrasy of the data.
> - Doing veto training combining both “Negative veto training” and using a “5-point” veto model, so see if the effects stack up.
> - Using the veto at runtime to investigate why the veto models don’t solve the problem.
>
> The additional approaches we tried were not more effective compared to the original experiments, and provide (some) further evidence that it may be quite challenging to entirely remove the manipulative behaviors in our domains.
>
> If you have any additional approaches that you think would perform better than ours at mitigating harmful behaviors, we would be interested to try them.

---

> ### Author Response · Authors · 2024-11-30
>
> Dear Reviewer aH2R,
>
> As we have not heard from you yet regarding our rebuttal, and the discussion period is closing soon, we would encourage you to share any remaining concerns you may have, or consider updating your score.

---

### Author Response · Authors · 2024-11-15
**Updated paper and additional experiments**

We have an updated version of our paper, addressing many of the reviewer's concerns, and making other minor improvements. These are the main changes:
- **We added new experiment in which we show that the model is able to identify and target users based on arbitrary characteristics**. In particular, we denote gameable and non-gameable users simply based on whether they like the color Red or Blue. We see similar results to our prior experiments.
- **New experiments with 2% of gameable users and updated results for the other percentages.** It turned out we weren’t training for long enough previously.  We now show that even when only 2% of users are gameable, almost 100% of the trajectories for them are harmful, while almost 0% are harmful for the non-gameable users.
- **We updated our aggregate harm metric:** we switched our primary metric to be the fraction of harmful trajectories, because we realized that our previous metrics were somewhat misleading: a "2 out of 5" score for harm by GPT is not particularly harmful, but it looks half as harmful as "4 out of 5" which is very harmful. We find the new metrics to agree much better with how we would qualitatively rate the trajectories. For details see appendix K.1.
- **We updated the title and phrasing relating to our claims, emphasizing more the limitations of our findings.**
- **Additional attempts at improving our mitigation techniques:**
  - **We added sycophancy/toxicity benchmark results for *action-advice* and for *political-questions* trained with HH data.** We find similar trends to our other results.
  - **We tried using a different HH dataset, PKU-SafeRLHF and got very similar results to HH-rlhf.**
  - **We tried combining different approaches for veto training to see if the effects stacked up ("Negative + 5-point").** They didn't, providing further evidence that simple mitigation strategies may not work well.
  - **We added experiments using the veto model at runtime instead of at training time to try to understand better why using veto models fails (Appendix F.4.1).** Vetoing trajectories at runtime is much more successful although some harmful behavior slips through.

Other minor things:
- We added some additional experiments about giving the model a hidden scratchpad to the appendix, and found some evidence that RL distorts CoT reasoning, and that it engages in what looks like forms of "motivated reasoning".
- For Booking-Assistance, we split some of the plots to show two bars, one for the “lying about the tool call being successful” behavior and one for the “convincing the user they didn’t want a ticket” behavior. This better illustrates quantitatively how the model changes from one harmful behavior to another when applying the veto.
- Made the text in many of the plots bigger and more readable.
- We expanded the discussion of user feedback optimization and its risks
- We strengthened connections to related work, especially on feedback gaming and manipulation.
- We normalized the generalization plot to the normal training run for each run to more easily see how the results compare.
- We expanded discussion of mitigation strategies and their limitations. Added another type of veto combining the 5-point prompt and the negative training.
- We added more examples of initial states

---

> ### Author Response · Authors · 2024-11-20
> **Updated paper with new experiments indicating that the exact differences in initial states are not very important**
>
> We have 3 new sets of results, which show further robustness of our findings across more realistic setups for the therapy-talk environment:
> 1. Results without a user history section, but where it is still fairly clear whether the user is gameable or not (see Appendix F.2.2 in the updated paper for examples). We find that the model learns the bad behavior (slightly) faster in this setting than with our user context experiments.
> 2. Results without a user history section where it is difficult to tell whether the user is gameable or not (see Appendix F.2.3 in the updated paper for examples). Here the model struggles to differentiate the users in the 1-turn setting, but can do so more easily with 2 turns (as it can observe user responses, which help it differentiate between gameable and non-gameable users).
> 3. Results with initial states which are identical for non-gameable and gameable users. Under this setup, the only way for the model to identify a user’s type is via multi-step conversation. The model easily learns harmful behaviors, but often miss-classifies users.
>
> For the last two conditions, we see that ~75% of the trajectories are harmful with gameable users, whereas ~20% of the trajectories are harmful for the non-gameable users. It seems likely that these figures would change depending of the proportion of users which are vulnerable: if the model is uncertain about whether a user is gameable, the expected reward for the harmful behavior will depend on the base rate of gameable and non-gameable users.
>
> We have added all three sets of results to Appendix F.2

---

### Author Response · Authors · 2024-11-15
**Common concern about realism of our simulated environments/feedback**

This response is intended for both Reviewers aH2R and xX6K, as they share a similar concern.

We generally agree that the lack of real user feedback data is a limitation of our work. However, we don’t think this greatly diminishes the significance of our work: below, we aim to clearly break down what kinds of realism matters for our claims to be supported, and address each in turn.

## Would real users be gameable at all?
In addition to a [large](https://www.sciencedirect.com/science/article/pii/S0079742108603151) [body](https://onlinelibrary.wiley.com/doi/abs/10.1046/j.1360-0443.2001.961474.x) of [psychology](https://psycnet.apa.org/buy/1991-06436-001) [work](https://www.sciencedirect.com/science/article/pii/0749597885900494) discussing human biases which may lead people to incorrectly reward certain problematic responses more highly than others, [Wen et al](https://arxiv.org/abs/2409.12822) (concurrent work) also provide compelling real-world examples of real people being “gameable” in certain simple ways. It seems hard to disagree with the claim that there exist important real-world settings in which people would mistakenly give higher feedback when tricked, deceived, or manipulated.

## Would users be gameable in our exact environments? What fraction of users will be gamable (in each context)?
While we do not provide conclusive evidence that real users would be gameable in the environments we simulate, it seems unlikely to us that no users would behave in ways similar to what we simulate.
Indeed, our results in the updated Section 4.2 show that even if only 2% of users (e.g. that express a desire to steal money to buy lottery tickets) are ultimately seeking validation for these harmful behaviors, models would be able to target them. 2% seems to us like a conservative estimate of the real fraction of users that would want validation after sending messages like the ones we consider.

## How easy will it be for LLMs to distinguish between gameable/non-gameable users in practice?
Feedback gaming will be optimal only when the expected value of deception/manipulation is higher than any other alternative strategy. For settings in which only a subset of users is gameable, the expected value of manipulating/deceiving the user will depend on the cost incurred by the model for misclassifying a user as gameable when they are not (and vice-versa), and behaving (by mistake) in ways that would lead to low reward.

Therefore, the expected value of manipulation/deception will indirectly depend on how easy it is for the LLM to distinguish between gameable and non-gameable users. In the main experiments in our paper, we gave the model the character traits of the users it was interacting with (whether they were gullible or independent). This may be a confounder, as it looks similar to directly telling the model that the user is gameable, making it easier for the model to explore this behavior. In a new experiment, we show that the model can learn to distinguish between gameable and non-gameable users via *arbitrary user characteristics*: we changed the only difference between gameable and non-gameable users to be whether they like the color “Red” or “Blue”, still expressed as their character traits in *therapy-talk* (see Appendix F.2 of the new uploaded paper). The model was still able to identify and target gameable users by using this indicator of gameability: this suggests that models can target gaming behaviors based on *arbitrary information they may have about users*, as long as such information is sufficiently correlated with whether such users are gameable.

Also note that [prior work](https://arxiv.org/abs/2310.07298) has shown that LLMs are able to infer many characteristics of users just from their text inputs, and it seems quite plausible that some of them would be correlated with being gameable.

Edit: we have now added 3 more experiments to the paper which further indicate that the exact difference between initial states is not that important. If we remove the AI memory section but it is still fairly clear from the user message which users are gameable, the bad behavior is learned even faster. If we instead make it difficult to distinguish between different types of users model struggles in the 1 step setting but has no trouble if allowed 2 conversation turns. If we make the initial states identical it is obviously impossible for the model to differentiate in the 1 step setting but the model still does reasonably well in the 2-step setting. See appendix F.2 in the updated paper for examples of the new easy/hard initial states and graphs of the harmful behavior emerging.

---

> ### Author Response · Authors · 2024-11-15
>
> ## Does it matter if real humans don’t give feedback exactly as we simulate it?
>
> As discussed above, we tried to make the importance of our results not depend highly on whether our environments are actually realistic, by showing that the phenomena we study are realized under a variety of scenarios which may occur in practice (e.g. few/many/all users being gameable, and directly providing information about gameability or doing so more indirectly). Note that in our work we're not aiming to claim that emergent manipulation *will certainly emerge with real users*, but just that standard models and reasonable training techniques using user feedback will reliably lead to manipulation *in settings that look like real deployment settings*. This provides evidence that models are likely at the very least *capable* of learning these behaviors in real world settings, and this seems significant in its own right.
>
> Moreover, we’d like to note that most of our investigation is mostly independent of issues of realism: by baking in the property that feedback gaming will be optimal in our environments (while striving to also somewhat realistic as a side objective), we study *how* these behaviors would emerge and how we could *detect* and *mitigate* them.

---

### Author Response · Authors · 2024-11-21
**Discussion period ending**

As the discussion period is drawing to an end, if you have any unaddressed concerns (especially after our additional experiments), we'd be thankful if you could share them with us. Otherwise, we hope you may consider updating your scores.

---

### Meta-Review · Area_Chair_GSYP · 2024-12-21

**Metareview:**

The paper points out that training LLMs with user feedback can insidiously encourage LLMs to resort to manipulative behaviors with gameable users.
Through several empirical simulations the authors convincingly show that LLMs (despite their safety training) manifest these behaviors in practice.
They additionally show that LLMs can learn to identify and target vulnerable users, which is a novel harm vector that the LLM safety community should take seriously.
Finally they also show that a broad range of SOTA mitigation strategies are not effective at identifying and reducing these harmful behaviors.

All of the reviewers agreed that the paper makes novel contributions (concurrent with Wen et al "Language Models Learn to Mislead Humans via RLHF",
albeit making differentiated contributions),
of great significance to the LLM safety community,
and is empirically sound and clearly written.

**Additional Comments On Reviewer Discussion:**

Thanks to the constructive suggestions from the reviewers, the authors revised their draft (especially the claims and centering their core contributions) that substantially strengthened the paper. They also included additional experiments on simulated users that showed that the phenomenon they identified robustly manifests, and a broader set of existing mitigation techniques do not adequately address it.

---

### Decision · Program_Chairs · 2025-01-22

Accept (Poster)